# Detecting, mapping, and suppressing the spread of a decade-long *Pseudomonas aeruginosa* nosocomial outbreak with genomics

William Stribling[1], Lindsey R Hall[1], Aubrey Powell[1,2], Casey Harless[1], Melissa J Martin[1], Brendan W Corey[1], Erik Snesrud[1], Ana Ong[1], Rosslyn Maybank[1], Jason Stam[1], Katelyn V Bartlett[1], Brendan T Jones[1], Lan N Preston[1], Katherine F Lane[1], Bernadette Thompson[3], Lynn M Young[3], Yoon I Kwak[1], Alice E Barsoumian[4], Ana Elizabeth Markelz[4], John L Kiley[4], Robert J Cybulski[1,5], Jason W Bennett[1], Patrick T Mc Gann[1]*, Francois Lebreton[1]*

[1]Multidrug-Resistant Organism Repository and Surveillance Network (MRSN), Walter Reed Army Institute of Research, Silver Spring, United States; [2]Department of Pathology, Brooke Army Medical Center, Joint Base San Antonio-Fort Sam Houston, San Antonio, United States; [3]Infection Prevention & Control, Brooke Army Medical Center, Joint Base San Antonio-Fort Sam Houston, San Antonio, United States; [4]Infectious Disease Service, Department of Medicine, Brooke Army Medical Center, Joint Base San Antonio Fort Sam Houston, San Antonio, United States; [5]Bacterial Disease Branch, Walter Reed Army Institute of Research, Silver Spring, United States

**\*For correspondence:**
patrick.t.mcgann4.civ@health.mil (PTMG);
francois.lebreton.ctr@health.mil (FL)

**Competing interest:** The authors declare that no competing interests exist.

## eLife Assessment

This **important** work presents an example of how genomic data can be used to improve understanding of an ongoing, long-term bacterial outbreak in a hospital with an application to multi-drug resistant *Pseudomonas aeruginosa*, and will be of interest to researchers concerned with the spread of drug-resistant bacteria in hospital settings. The **convincing** genomic analyses highlight the value of routine surveillance of patients and environmental sampling and show how such data can help in dating the origin of the outbreak and in characterising the epidemic lineages. These findings highlight the importance of understanding environmental factors contributing to the transmission of *P. aeruginosa* for guiding and tailoring infection control efforts.

**Abstract** Whole-genome sequencing is revolutionizing bacterial outbreak investigation, but its application to the clinic remains limited. In 2020, prospective and retrospective surveillance detected a *Pseudomonas aeruginosa* outbreak with 253 isolates collected from 82 patients in 26 wards of a hospital. Its origin was dated to the late 1990s, just after the facility opened, and patient-to-patient and environment-to-patient cases of transmission were inferred. Over time, two epidemic subclones evolved in separate hosts and hospital areas, including newly opened wards, and hospital-wide sampling confirmed reservoirs persisted in the plumbing. Pathoadaptive mutations in genes associated with virulence, cell wall biogenesis, and antibiotic resistance were identified. While the latter correlated with the acquisition of phenotypic resistances to first (cephalosporin), second (carbapenems), and third (colistin) lines of treatment, maximum parsimony suggested that a truncation in a lipopolysaccharide component coincided with the emergence of a subclone prevalent in long-term infections. Since initial identification, extensive infection

control efforts guided by routine, near real-time surveillance have proved successful at slowing transmission.

## Introduction

*Pseudomonas aeruginosa* is a versatile, opportunistic human pathogen causing a wide range of infections of both community and hospital origin (*Kerr and Snelling, 2009*; *September et al., 2007*; *Rodríguez-Rojas et al., 2012*). In the clinic, patients with cystic fibrosis (CF), severe burns, or neutropenia are of highest risk, and cases are associated with high morbidity and mortality (*Rodríguez-Rojas et al., 2012*). Furthermore, the management of these infections is increasingly difficult due to the global emergence of multidrug-resistant (MDR) strains, especially carbapenem-resistant *P. aeruginosa* (CRPA), which now account for 16–30% of cases in the USA (*World Health Organization, 2017*). In response, global and national agencies have categorized CRPA as a 'critical' pathogen posing a serious threat to public health (*World Health Organization, 2017*).

While high-risk MDR lineages, exemplified by the most globally prevalent sequence type (ST) 235, are enriched in isolates carrying acquired carbapenemases and/or extended-spectrum β-lactamases (ESBLs) (*Hong et al., 2015*; *Del Barrio-Tofiño et al., 2020*), the hallmark of *P. aeruginosa* lies in its remarkable propensity for developing resistances through chromosomal mutations (*Blair et al., 2015*). Within the hundreds of intrinsic genes comprising this mutational resistome, several harbor well-characterized modifications that are the most common cause of resistance for nearly all classes of drugs (*Blair et al., 2015*). These include gain-of-function mutations in gyrase subunits GyrAB (fluoroquinolones), penicillin-binding protein PBP3 (cephalosporins), various Mex efflux systems (multiple antimicrobials), as well as inactivation of the porin OprD (carbapenems) or the sensor component of the PhoPQ system, the latter associated with resistance to last-line polymyxin antibiotics (*Li et al., 2012*; *Bruchmann et al., 2013*; *Clark et al., 2019*).

Being capable of both acute and chronic infection, *P. aeruginosa* possesses a large arsenal of virulence factors such as a type 4 pili (T4P), O-antigens and lipopolysaccharides (LPS), extracellular enzymes, exotoxins (e.g. ExoS/U), a flagellum and five types of secretion systems (TSSs) (*Jurado-Martín et al., 2021*). These are generally part of complex regulatory pathways in which two-component systems (TCSs) often play a major role (*Jurado-Martín et al., 2021*; *Juan et al., 2017*). The ability of *P. aeruginosa* to form biofilm is a key aspect of its persistence in patients with chronic infections, as well as in environmental niches (*Thi et al., 2020*). In the hospital, biofilms are often found in water-related environments, and eradication is particularly difficult due to increased disinfectant resistance (*Kerr and Snelling, 2009*; *Smith and Hunter, 2008*). This, in turn, poses a challenge for infection prevention and control (IPAC) as illustrated by the increasing reports of *P. aeruginosa* outbreaks and hospital-acquired infections (HAI) linked to contaminated sinks, drains, or taps (*Naze et al., 2010*; *Quick et al., 2014*; *Sukhum et al., 2022*; *Parcell et al., 2018*).

Here, routine genome-based surveillance of MDR clinical isolates across the US Military Health System (MHS) identified an epidemic cluster of *P. aeruginosa* in a single hospital in 2020. Retrospective analysis of isolates from the preceding 10 years, together with Bayesian phylogenetics, revealed a protracted outbreak clone with cases from all hospital floors and an origin dated to the late 1990s. Dissection of this unique dataset sheds new light on the evolution, persistence, and routes of transmission of *P. aeruginosa* in the clinic.

## Results

### *P. aeruginosa* ST-621 is endemic in a large US tertiary hospital

The Multidrug-Resistant Organism Repository and Surveillance Network (MRSN) performed whole-genome sequencing (WGS) of 5129 *P. aeruginosa* isolates received between 2011 and 2020 from 71 facilities around the globe (*Lebreton et al., 2021*). While a total of 547 STs were represented, high-risk epidemic lineages ST-235/*exoU+* (10%) and ST-244/*exoS+* (5%) were the most prevalent and widespread with isolates recovered from 37 and 28 facilities, respectively (*Figure 1A*). By contrast, isolates from lineage ST-621/*exoS+* also represented 5% of the total STs, but originated from just 6 facilities, with 89% from a single US tertiary hospital (Facility A) and sharing high-level genetic relatedness (0–23 allelic differences) by core genome multilocus sequence typing (cgMLST) (*Figure 1B*). Furthermore,

**eLife digest** Many bacteria thrive in moist hospital environments, such as sinks, drains and medical equipment, and pose a significant infection risk. For example, the bacterium *Pseudomonas aeruginosa* can cause severe infections in patients with burns, wounds or weakened immune systems. This species can further develop resistance to many antibiotics, making infections more difficult to treat and control.

Hospitals use infection prevention programs to detect and limit bacterial spread. But identifying how and where the bacteria persist remains a challenge. To reveal hidden outbreaks and identify environmental sources, scientists have started using genome sequencing – the process of reading a bacterium's complete genetic code – to monitor how hospital infections begin and propagate.

To underscore the value of this method, Stribling et al. investigated a decades-long *Pseudomonas aeruginosa* outbreak in a hospital in the United States, despite ongoing infection control efforts. By sequencing the genomes of more than 250 bacterial samples collected over ten years, the team traced the outbreak's origin to the late 1990s, soon after the hospital opened.

The same strain spread across multiple wards and evolved into two related subgroups. Genetic and environmental analyses revealed that contaminated sink drains acted as long-term reservoirs, repeatedly infecting patients even years apart. The bacteria gradually became resistant to several major antibiotics. Guided by these genomic insights, targeted infection control actions – such as sink decontamination and routine genomic monitoring –finally reduced new infections and helped contain the outbreak.

The findings of Stribling et al. demonstrate how routine genome sequencing can uncover hidden infection routes and guide more effective hospital control measures. Public health systems could use similar approaches to detect outbreaks earlier and prevent prolonged hospital contamination. Before this can become routine, hospitals will need access to affordable sequencing technologies, trained personnel and real-time data sharing. In the long run, such integrated genomic surveillance could protect patients and reduce the global burden of antibiotic-resistant infections.

---

of the 5 additional facilities from which ST-621 was detected, patient transfer from Facility A was observed in three instances, and isolates from the remaining two were genetically distinct strains (88–89 allelic differences) (*Figure 1B*).

Retrospective analysis revealed that a total of 253 ST-621 isolates were collected from Facility A (a 425-bed medical center and the only level 1 trauma center in the MHS) between May 2011 and December 2020. The majority were isolated from respiratory (42%) and urine (39%) samples, followed by wound (7%), surveillance (6%), tissue (3%), and blood cultures (2%) (*Supplementary file 1*).

## A protracted outbreak with cases dispersed throughout floors and wards

Phylogenetic analysis confirmed that all ST-621 isolates from Facility A belonged to the same outbreak clone with an average distance (all-vs-all pairwise comparison) of just 38 single nucleotide polymorphisms (SNPs) and an interquartile range of 19 (*Figure 2A*, *Figure 2—figure supplement 1*, and *Supplementary file 2*). A total of 82 patients (average age of 57.2±21.0 and male/female ratio of 1.7) were represented, with the majority (56%) contributing serial isolates. In particular, six patients (ID 10, 27, 34, 35, 49, and 63) showed prolonged carriage, best exemplified by patient 35 from whom 21 urine isolates were collected over 4.5 years (*Figure 2A and B*).

Two outbreak subclones, designated SC1 and SC2, were identified from the tree topology and showed increased genetic relatedness within each (average distance of 24 and 31 SNPs, respectively) (*Figure 2A*). Overall, ST-621 isolates were observed in 26 wards/units from all 7 floors of two buildings (Main and Tower Buildings) in Facility A (*Figure 2C*), yet subclones SC1 and SC2 were spatially and temporally distinct. 78% of SC1 isolates were collected from patients in the Main Building and 60% were detected before 2014. In contrast, 52% of SC2 isolates originated from patients in the Tower Building and 77% were collected after 2014 (*Figure 2A*). Specifically, the early spread of SC1 was largely connected to patients in adjacent ICUs and step-down units on the second and third floors (wards 1–10) of the Main Building. The subsequent emergence of SC2 largely centered in patients

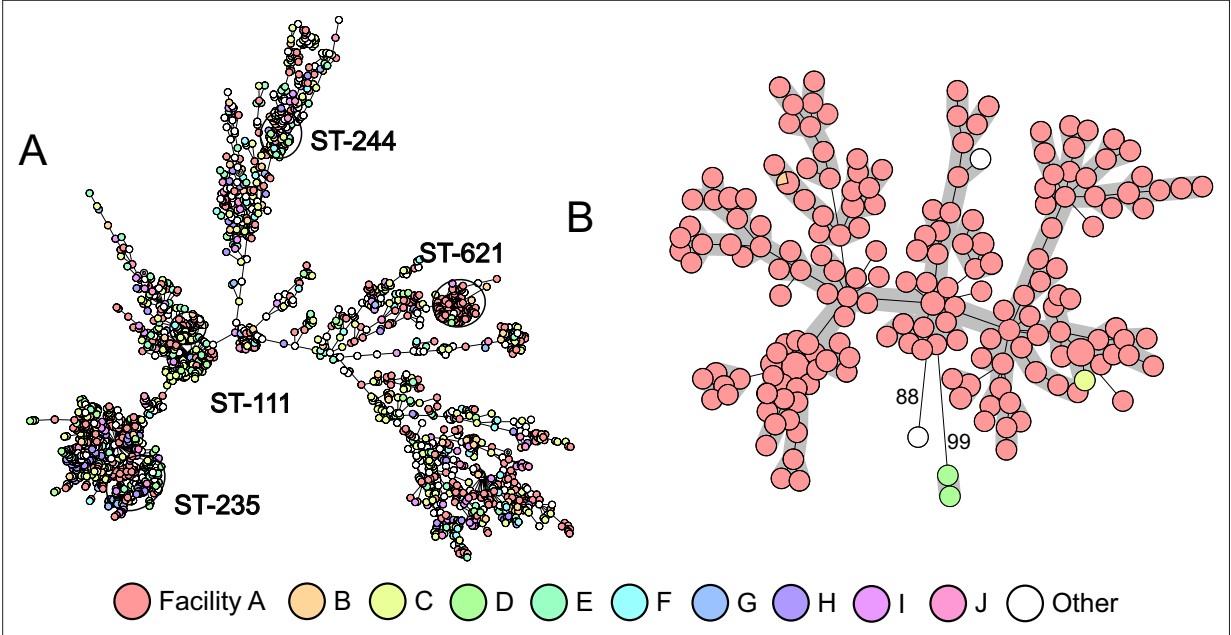

**Figure 1.** Minimum spanning trees of *P. aeruginosa* core genome multilocus sequence typing (cgMLST). Isolates are color-coded by facility, showing the 10 most prominent. (**A**) Includes all 5129 *P. aeruginosa* in the Multidrug-Resistant Organism Repository and Surveillance Network (MRSN's) collection from 2011 to 2020. Clusters of isolates belonging to ST-235, -244, and -621 are indicated. (**B**) Includes all ST-621 isolates from 2011 to 2020. Genetic distances higher than 23 allelic differences are indicated.

receiving care in adjacent ICUs (wards 20–21) on the third floor within the Tower Building (*Figure 2A and C*). Throughout the outbreak, new cases due to either SC1 or SC2 were also detected from patients in the emergency room (ER). Unlike the isolates from other locations, these ER isolates were nearly all (32% vs 92%, p<0.0001) from urine cultures and, as expected, the majority (72%) were serial isolates from returning patients at Facility A (*Figure 2C*, *Supplementary file 1*). The remaining samples were collected from 9 patients for whom a previous visit at Facility A is suspected but could not be established as no previous isolates were collected from them and no historical patient data was available.

The epidemic curve revealed that between 2011 and 2016, an average of 11.5 new cases was detected every year. From 2017 to 2021, only 12 new patients were identified (3 cases a year) (*Figure 2D*). Although the rate of outbreak cases appeared to decrease, the overall number of *P. aeruginosa* cases in Facility A (irrespective of lineages) remained stable throughout the studied period. However, due to staffing issues, the number of isolates received/sequenced by the MRSN every year was not consistent (*Figure 2D*, *Figure 2—figure supplement 2*).

### An origin dated soon after the opening of Facility A

Though the collection of clinical isolates at Facility A only started in 2011, genetic diversity was already evident (16.3 SNPs on average), suggesting an earlier origin (*Supplementary file 2*). Molecular clock phylogenetic analyses of all studied isolates inferred the time of the most recent common ancestor (MRCA) to be in 1999 (±3 years), soon after the opening of the Main Building of Facility A in 1996 (*Figure 3*). Furthermore, this analysis predicted that, despite the observed temporal succession in sequenced isolates, SC2 was not a descendant of SC1. Instead, the two subclones emerged concurrently with respective MRCAs dated in 2004 (±2 years) (*Figure 3*).

Despite this predicted early origin, it is only after facility expansion and the opening of the Tower Building in 2012 that the first SC2 isolates were ever sampled (*Supplementary file 1*). Specifically, the earliest SC2 isolate collected was from a wound infection in patient 27. Remarkably, the 15 serial isolates collected over 3 years (2012–2015) from this patient were not monophyletic, and their MRCA coincided with the emergence of this subclone (*Figure 3*).

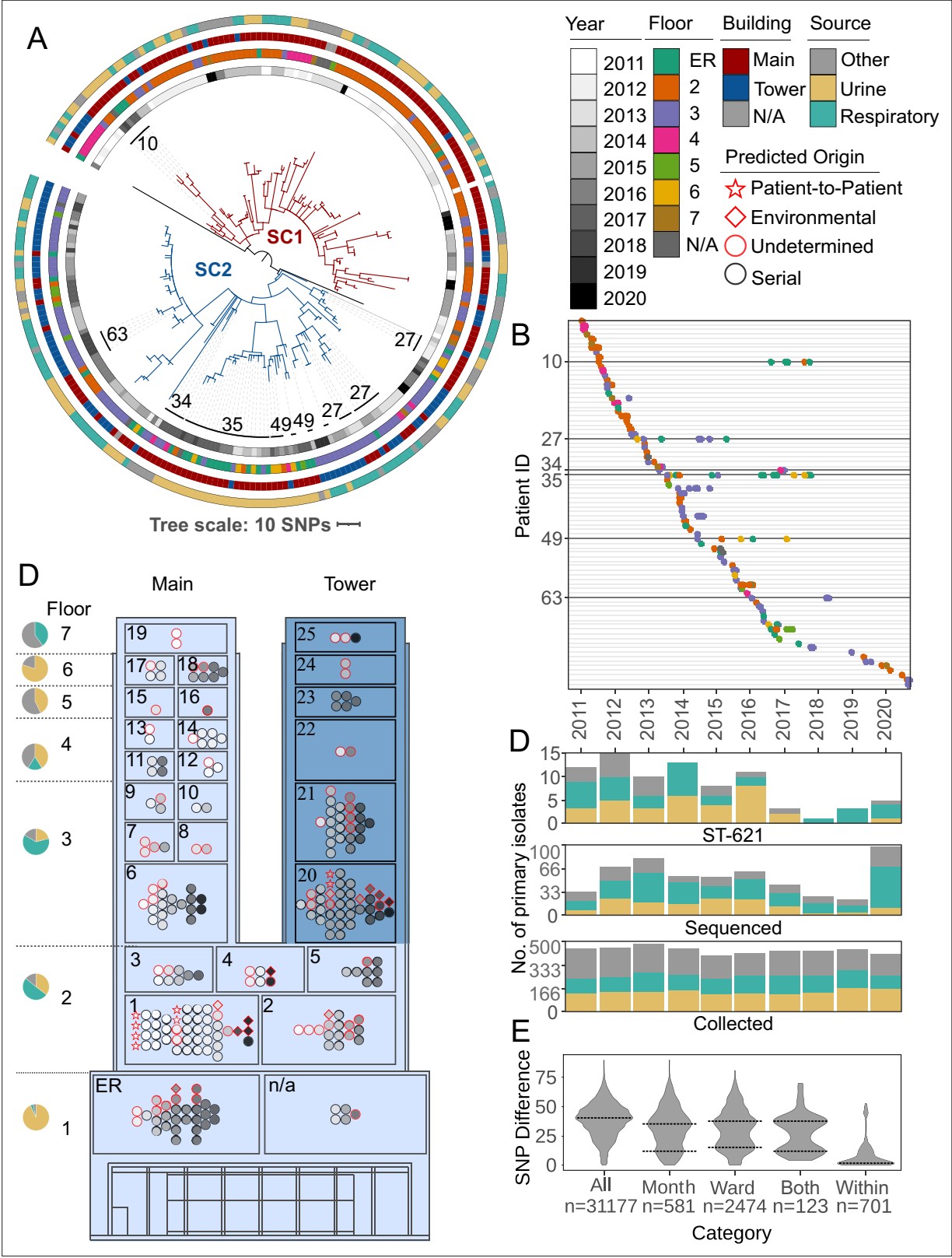

**Figure 2.** Phylogenetic and spatiotemporal analysis of 253 clinical ST-621 *P. aeruginosa* isolates. (**A**) Single nucleotide polymorphism (SNP)-based, core genome phylogeny. Outbreak subclones are colored with red (SC1) or blue (SC2) branches. From inner- to outermost, datasets show the year, the floor, the building, and the source of isolation. Patients with prolonged carriage (>1 year) are indicated. (**B**) Timeline of isolate collection. Each line indicates a single patient. Isolates are shown as dots colored by the floor the patient was on. Patients with prolonged carriage are highlighted. (**C**) Simplified

*Figure 2 continued on next page*

*Figure 2 continued*

floor plan of the Main (light blue) and Tower (dark blue) Buildings of Facility A. Wards and floors are numbered. For each floor, a pie chart indicates the proportion of isolates collected from urine, respiratory, or other cultures. Within each ward, isolates are colored by year of collection with a red outline indicating primary (first) isolates. Symbols indicate the presumed origin of infection. (**D**) Bar charts showing for each year the number of all *P. aeruginosa* primary clinical isolates collected at Facility A (bottom), the subset of genome-sequenced primary isolates (middle), and the subset of identified ST-621 isolates (top). Bars are color-coded by the isolation source. (**E**) Distribution of SNP distances for all inter-patient isolates or subsets collected in the same month, the same ward, or both. SNP distances within patient isolates are also indicated. Dotted lines indicate the local maxima.

The online version of this article includes the following figure supplement(s) for figure 2:

**Figure supplement 1.** Bar chart showing the distribution of single nucleotide polymorphism (SNP) distances for all inter-patient isolates.

**Figure supplement 2.** Bar charts showing the percentage of sequenced *P. aeruginosa* isolates compared to the total collected by Hospital A stratified by year (2011–2020) and isolate types (respiratory, urine, or other).

## Environmental reservoirs sustain nosocomial transmission

The presumed origin of an infection, either directly from patient-to-patient or from environment-to-patient, was designated based on SNP thresholds (starting from pairwise comparisons for all inter-patient isolates, *Supplementary file 2*) and known epidemiological links, or lack thereof. Indeed, while a unimodal distribution of SNPs (peak at 40.5) was observed for the whole cohort, a bimodal profile appeared when inter-patient isolates from the same month, the same ward, or both were selected (local maxima of the lower peak at 12.4, 15.3, and 11.6 SNPs, respectively) (*Figure 2E*). This pool of virtually identical isolates from patients with spatial and temporal overlap represented the most likely cases of direct, patient-to-patient transmission. Pairs of isolates from only 10 patients satisfied these

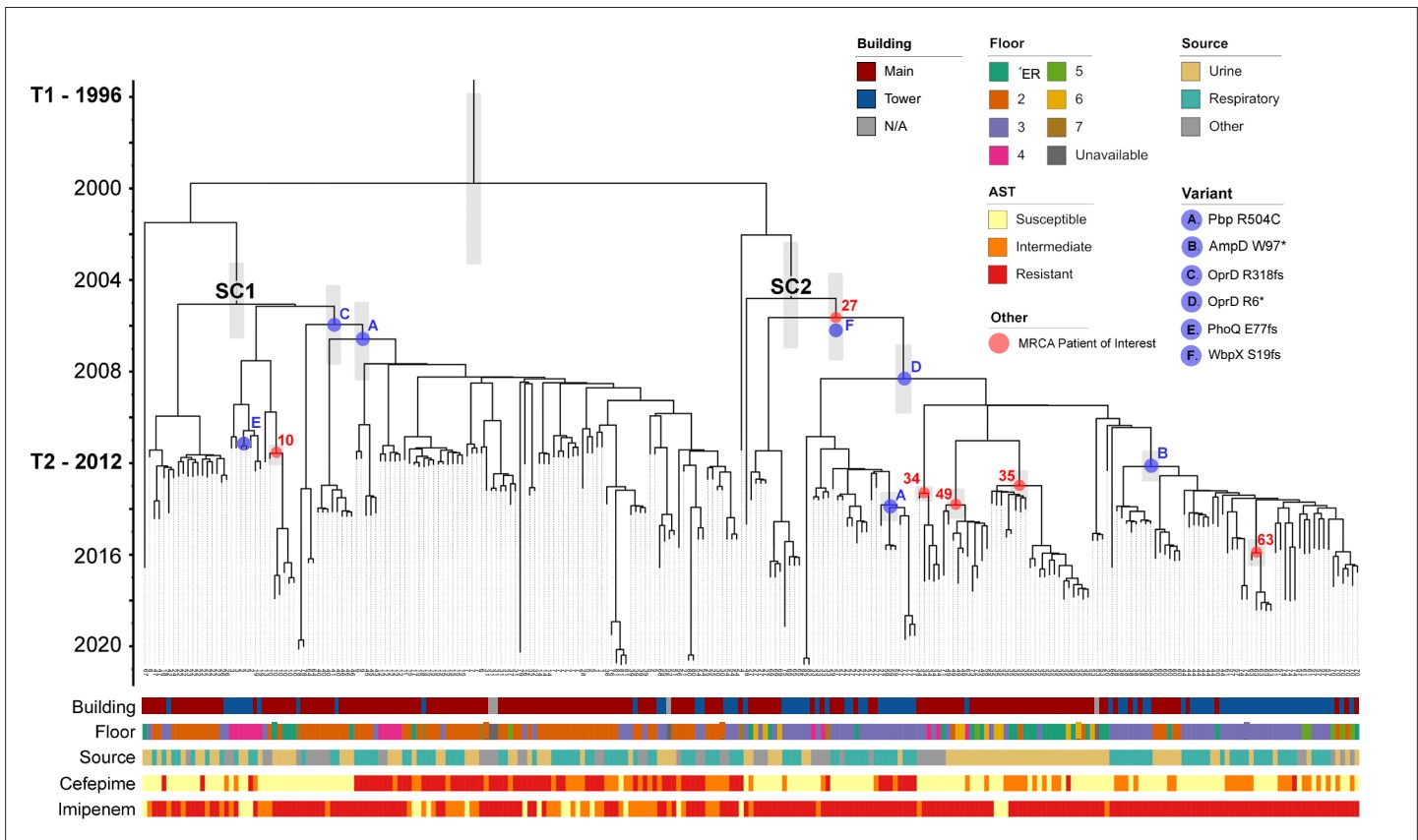

**Figure 3.** Dated phylogeny of 253 ST-621 outbreak isolates. Time-stamped phylogeny. Error bars for annotated nodes are shown as gray boxes. Years are indicated on the left, along with the opening of the Main Building (T1) and the Tower Building extension (T2). The most recent common ancestor (MRCA) of SC1 and SC2 is indicated in black text above the node. The MRCA of isolates from patients with prolonged carriage is shown in red, indicated by their patient ID. The predicted emergence of select variants is shown in blue. Data showing the floor and source isolates were collected from is shown, as in *Figure 1*. Select datasets are shown, including the antibiotic susceptibility testing results for cefepime (cephalosporin) and imipenem (carbapenem).

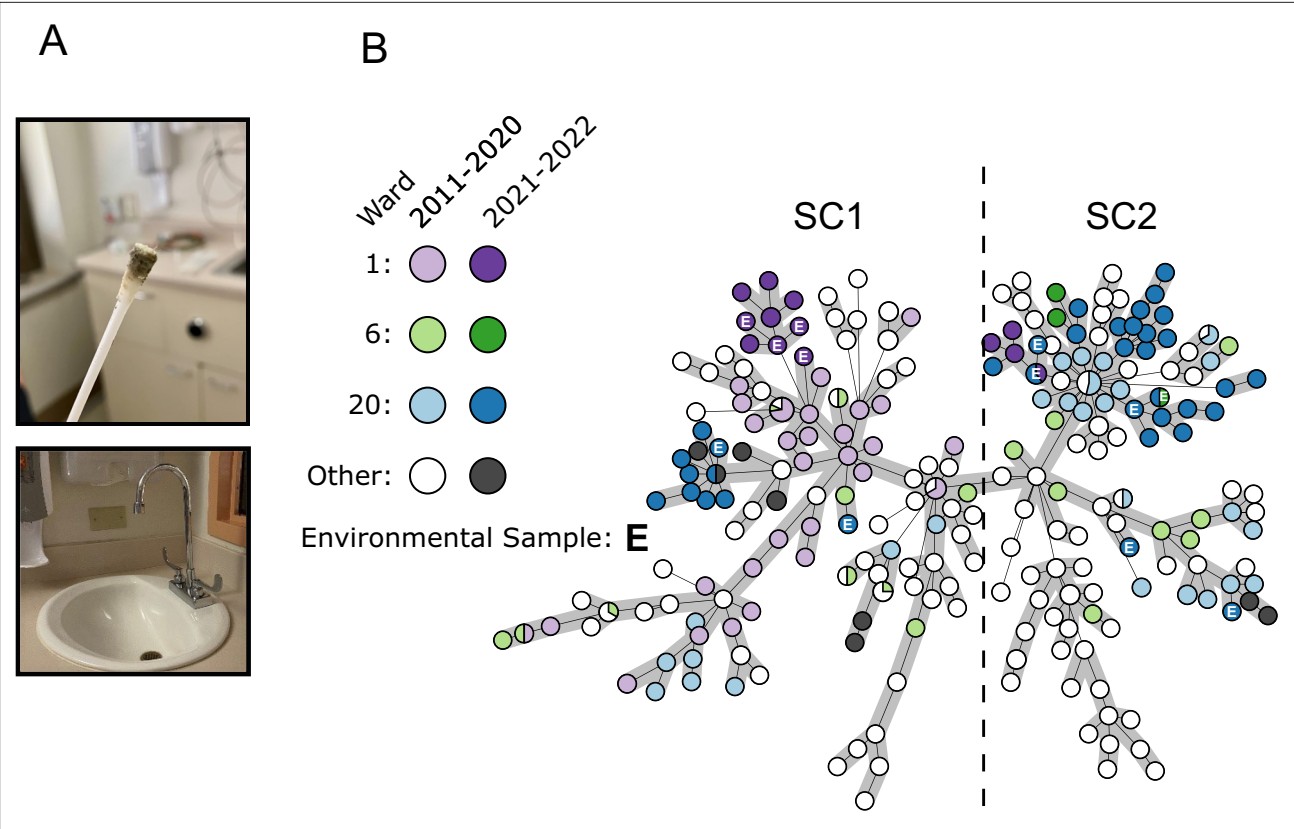

**Figure 4.** Identification of environmental reservoirs of the ST-621 outbreak clone in sink drains. (**A**) Photos depict a swab from a sink drain in a patient room and the typical sink design (shallow basins and gooseneck faucets) in Facility A. (**B**) Minimum spanning tree of 326 Facility A ST-621 *P. aeruginosa* isolates, including contemporary isolates collected in 2021 and 2022 from clinical (n=56) and environmental (n=17) sources. Isolates from wards #1, 6, and 20 where environmental contamination was identified are colored. Environmental isolates are indicated with the letter E.

conditions (*Supplementary file 1*): All carried subclone SC1, occurred in 2011 and 2012 (with two exceptions), and six were hospitalized in ICU #1 in the Main Building of Facility A (*Figure 2C*).

In contrast, a presumed environmental origin was designated when a new patient was infected with a virtually identical strain from a preceding patient in the same ward, but no known temporal overlap was established for at least 1 year (based on isolates collection dates). Twelve cases met these criteria (*Figure 2C* and *Supplementary file 2*). Among them, three patients (ID 57, 61, and 74) in ICU #20 on the Tower Building had primary isolates (first isolates from a patient) of subclone SC2 that were distinct by only 0–4 SNPs despite being collected years apart (in 2015, 2016, and 2018, respectively) (*Figure 2C*). Furthermore, an origin from contaminated surfaces was also predicted for seven of the nine most recent patients (2018–2020). This included five cases caused by the apparent resurgence of SC1 in ICUs #1–2 of the Main Building (*Figure 2C*).

Genomic inferences were supported by contemporary environmental sampling, conducted in 2021 and 2022, during which a total of 159 swabs were taken from 55 distinct locations in 8 wards/units, including the ER in Facility A. This confirmed the existence of reservoirs (i.e. sink drains located in patient rooms) of both SC1 and SC2 isolates (n=17) in seven distinct patient rooms in wards #1, 2, 6, and 20 throughout the Main and Tower Buildings of Facility A (*Figure 4*, *Supplementary file 3*). In that timeframe, an additional 56 clinical isolates (25 SC1 and 31 SC2) from 13 patients (including 11 new cases) were collected, indicating transmission events were still occurring (*Figure 4*).

## Selection on resistance genes and rising prevalence of CRPA due to porin defects

Phenotypically, a prototypical outbreak isolate showed non-susceptibility to aminoglycosides (except for amikacin), third and fourth generation cephalosporins, carbapenems, and fluoroquinolones

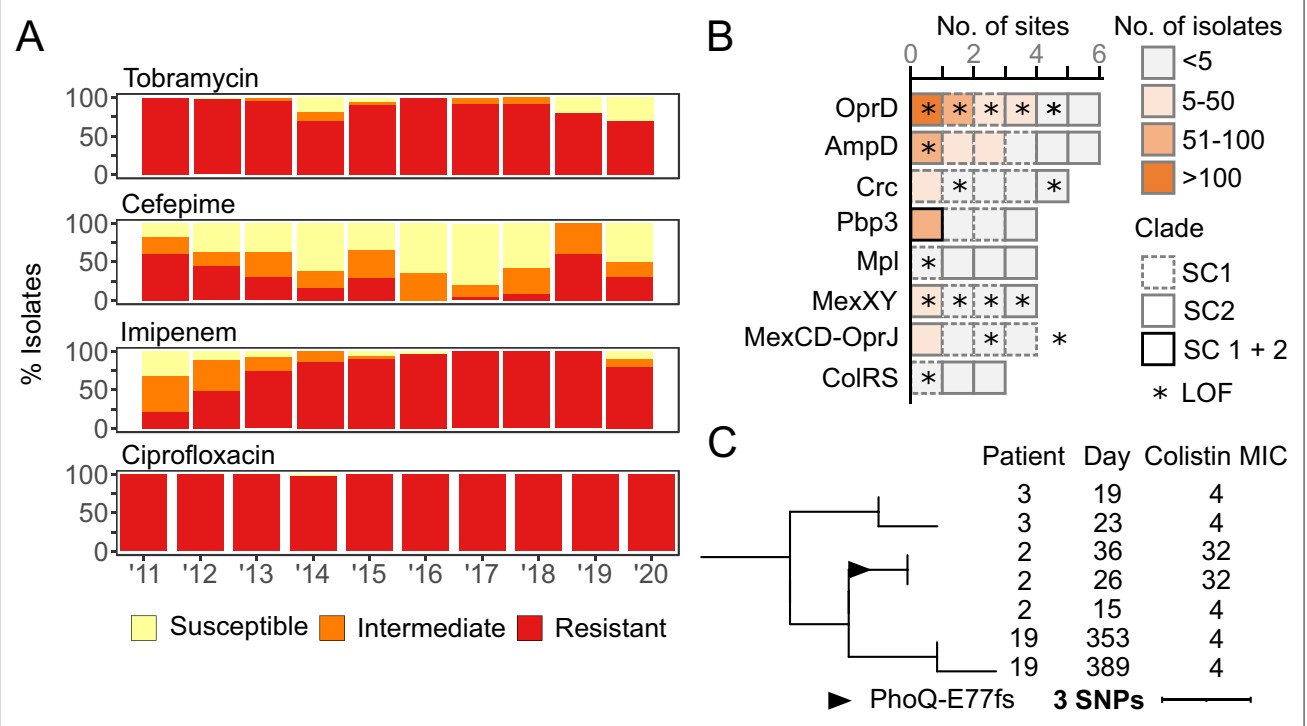

**Figure 5.** Antibiotic resistance and poly-variant sites of 253 outbreak isolates. (**A**) Bar charts showing the proportion of clinical isolates resistant to tobramycin, cefepime, imipenem, and ciprofloxacin each year from 2011 to 2020. (**B**) Chart of mutations (excluding synonymous) in resistance-associated genes. Each block indicates one distinct mutation observed within the corresponding gene. Blocks are colored by the number of isolates sharing this exact mutation. Blocks are outlined to indicate whether the mutation is found within SC1, SC2, or isolates from both subclones. Mutations causing a predicted loss-of-function (LOF) are indicated with a star. (**C**) Pruned single nucleotide polymorphism (SNP)-based phylogeny showing the acquisition of colistin resistance in Patient 2. Patient ID, day of collection (since the first outbreak isolate), and available colistin MIC are shown on the right. The most recent common ancestor (MRCA) of the PhoQ E77fs mutation is indicated with a triangle.

(*Supplementary file 4*). However, resistance profiles were not homogenous, and temporal patterns were observed (*Figure 5A*). In particular, while 63% of the isolates collected in 2011 were non-susceptible to imipenem, this fraction gradually increased to >95% by 2014 and beyond. In the same timeframe, the prevalence of isolates with non-susceptibility to cefepime decreased from 81% to 37% (*Figure 5A*). Both phenomena were linked to the decline of SC1 and rise of SC2, with the latter displaying significantly higher (98–76% in SC1, chi-square=53.5, and p<0.001) and lower (38–66%, chi-square=25.3, and p<0.001) rates of resistance to imipenem and cefepime, respectively (*Figure 3*). In contrast, resistance to tobramycin and ciprofloxacin was highly prevalent and remained relatively stable over the sampling period (*Figure 5A*).

Genotypically, outbreak isolates contained no plasmids (based on the finished genomes of a diverse subset of isolates, *Supplementary file 1*) and, besides an intrinsic AmpC-type β-lactamase, only carried two acquired resistance genes, known to confer resistance to ciprofloxacin (*crpP*) and tobramycin (*aac(6')-Ib4*) (*Supplementary file 1*). To explain other phenotypic resistances, and in the absence of an acquired ESBL or carbapenemase, a genome-wide analysis of non-synonymous (NSY) and loss-of-function (LOF) mutations was performed for all outbreak isolates (*Supplementary file 5*). From a list of 164 genes (*Supplementary file 6*) for which known chromosomal mutations have been linked to antibiotic resistance (*Jurado-Martín et al., 2021*; *Juan et al., 2017*; *Thi et al., 2020*), 68 distinct mutations (including 20 predicted LOF) in 37/164 genes were identified in at least one outbreak isolate (*Supplementary file 7*). Of particular interest, eight genes (or functional clusters) were each mutated independently three or more times (*Figure 5B*).

The most frequently mutated gene was *oprD*, encoding a porin that facilitates the diffusion of amino acids, small peptides, and carbapenem antibiotics into the cell (*Li et al., 2012*). Five of the six independent mutations identified in OprD were predicted LOF, including an R6* stop codon and an R318fs frameshift observed in 115 SC2 and 92 SC1 isolates, respectively (*Figure 5B*, *Supplementary*

*file 7*). Phylogenetic analysis using parsimony (PAUP) predicted that both the R318fs and R6* mutations occurred early (between 2006 and 2008) within the evolution of their respective subclones (*Figure 3*). Despite this early origin, highly carbapenem-resistant SC2-OprD-R6* isolates were first sampled from patients in mid-2012 and only became predominant from 2014 onward (*Figure 3*).

Among all other mutations in antibiotic resistance-associated genes (*Supplementary file 7*), only two additional variants were found in >50 outbreak isolates: a W97* stop codon in β-lactamase expression regulator AmpD and a R504C substitution in penicillin-binding protein PBP3 (*Figure 5B*). The LOF in AmpD, predicted to have emerged in 2012, was observed in a subset of 51 monophyletic SC2-OprD-R6* background isolates that were not consistently resistant to cefepime (*Figure 3*). In contrast, the R504C substitution in PBP3 was acquired independently twice (within SC1-OprD-R318fs in 2007 and within SC2-OprD-R6* in 2014) and was strongly associated with a resistance to fourth-generation cephalosporins (*Figure 3*). Overall, *ampD* and *pbp3* were among the genes with the most independently acquired NSY or LOF mutations, with six and four distinct sites, respectively (*Figure 5B*). Accumulation of mutations, albeit in small numbers of isolates, was also observed in other genes involved in β-lactam resistance (e.g. catabolite repression control *crc* and *mpl* UDP-muramic acid/peptide ligase), as well as antibiotic efflux systems (*mexXY* and *mexCD-oprJ*) (*Figure 5B*).

## Sporadic emergence of colistin resistance

Though rare, the presence and impact of mutations in genes associated with resistance to colistin, a last-line antibiotic, was examined. NSY mutations were detected in the TCSs ColRS (three distinct sites in four isolates) (*Figure 5B*) and PmrAB (two distinct sites in three isolates) (*Supplementary file 7*), but colistin MICs were the same as wild-type isolates (4 mg/L). By contrast, Patient 2 (who had a history of colistin therapy) acquired a predicted LOF mutation (E77fs) in sensor protein PhoQ in two serial isolates resulting in an eightfold increase in MIC (32 mg/L) (*Figure 5C*). Notably, the emergence of this colistin-resistant strain, which occurred in early 2011 (*Figure 3*), remained constrained to a single patient, and no further spread was detected.

## Mutational convergence in virulence, cell wall biogenesis, and signaling pathways

Genome-wide, and excluding synonymous mutations, 64 genes were each mutated independently two or more times over the course of this outbreak (*Supplementary file 5*). Compared to the distribution observed for the whole genome, these 64 genes were functionally enriched for signal transduction (28% vs 6% in the whole genome, chi-square, p<0.001) and cell wall/membrane biogenesis (17% vs 7%, p=0.0017) (*Figure 6A*). For the latter category, besides genes already discussed for their direct role in antibiotic resistance (i.e. *pbp3*, *oprD*, *mpl*, and *mex* loci), six other genes were recurrently mutated in outbreak isolates, including LPS biosynthesis *rmlA* and choline transporter *betT2* (*Figure 6A*).

For genes associated with signaling functions, various TCS involved in pathogenesis were represented, including *narXL*, *fleRS*, and the catabolic regulation *cbrAB* (*Figure 6A*). Notably, *cbrAB* was the most independently mutated loci in this dataset with 16 distinct NSY mutations in 80 isolates from both subclones. Beyond TCS, sigma factor *pvdS*, required for toxin expression (6 NSY and 1 LOF), motility regulator *morA* (11 NSY and 2 LOF), and mucoidy regulatory system *mucAB* (4 sites including three predicted LOF) were also recurrently mutated (though rarely fixed) in outbreak isolates (*Figure 6A*).

Finally, at the level of functional pathways, and compared to the theoretical frequency for an even distribution of mutated sites across the whole genome (0.11 sites per kb of coding sequence), NSY and LOF mutations were significantly enriched in genes involved in the biosynthesis of the Type IV secretion system (0.59 sites/kb, chi-square, p<0.001), flagellar biosynthesis and chemotaxis (0.21 sites/kb, p=0.0264), alginate production (0.31 sites/kb, p=0.0088), and O-antigen production (0.36 sites/kb, p=0.0126) (*Figure 6B*, *Supplementary file 8*). Importantly, the WbpX D-rhamnosyltransferase, part of the O-antigen biosynthesis pathway, was disrupted (S19fs) in 127 SC2 isolates (*Supplementary file 5*). PAUP predicted this frameshift occurred in 2005 (±24 months), concomitant with the predicted emergence of SC2 (*Figure 3*).

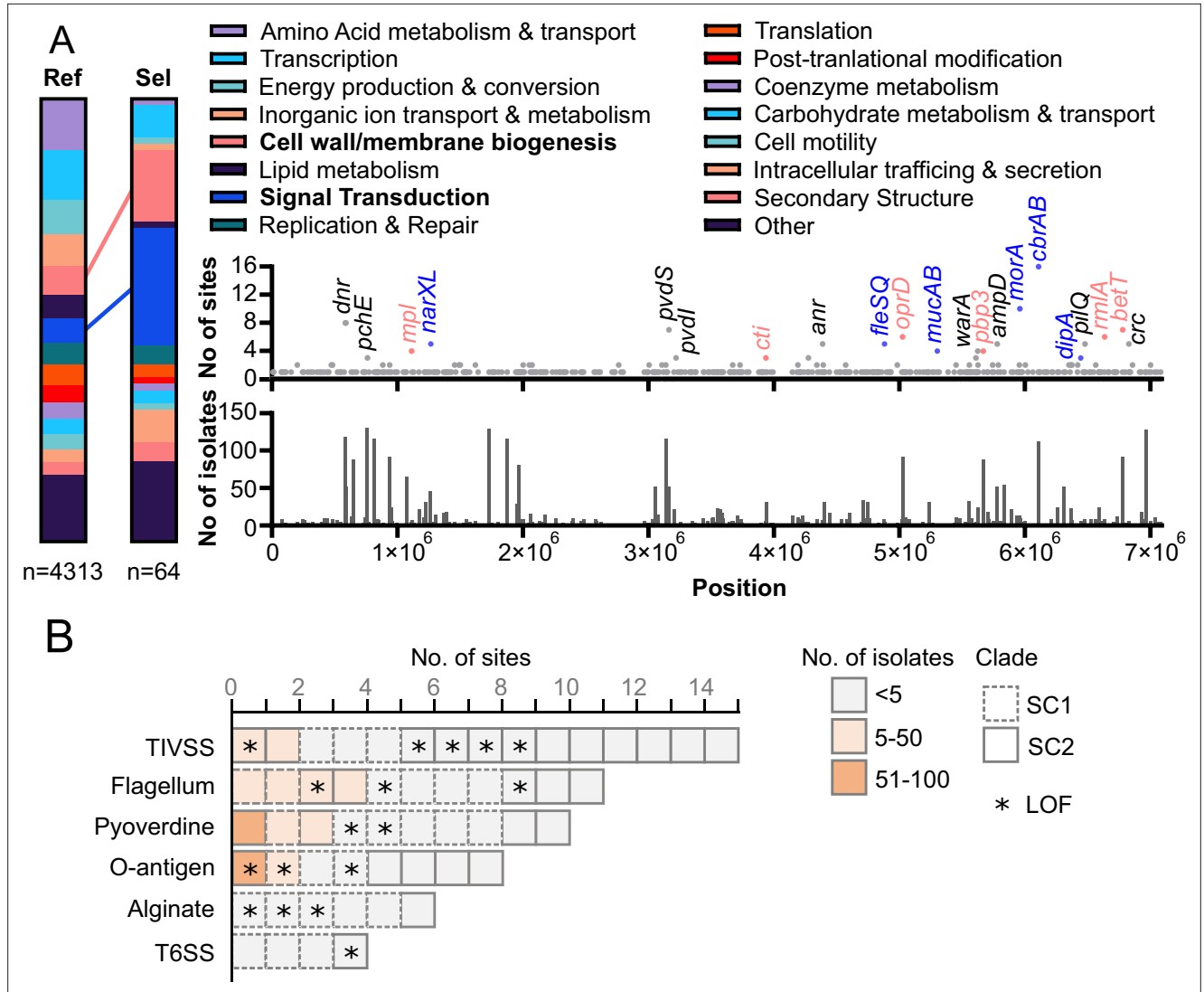

**Figure 6.** Identification of pathoadaptive mutations. (**A**) Distribution of COG functional categories for all genes (Ref) compared to the subset of genes (Sel) with two or more distinct mutated sites (excluding synonymous) in outbreak isolates. Significant enrichment in cell wall biogenesis and signal transduction categories is indicated. A positional chart of mutations across the entire chromosome of a closed reference genome for isolate 4605 is provided. The number of unique mutation sites within each gene is indicated (top), as well as the number of outbreak isolates which carried each mutation (bottom). Genes with three or more distinct mutated sites are labeled and genes involved in cell wall biogenesis and signal transduction genes are color-coded. (**B**) Accumulation of mutations (excluding synonymous) in genes and pathways associated with *P. aeruginosa* pathogenesis. Each block indicates one distinct mutation observed within a gene part of the indicated functional group (data available in ***Supplementary file 8***). Blocks are colored by the number of isolates sharing this exact mutation. Blocks are outlined to indicate whether the mutation is found within SC1, SC2, or isolates from both subclones. Mutations causing a predicted loss-of-function (LOF) are indicated with a star.

## Discussion

WGS technology is revolutionizing IPAC. Here, routine surveillance of MDR pathogens across the US MHS was critical in uncovering a decades-long *P. aeruginosa* epidemic cluster. With traditional approaches alone (***Giani et al., 2018***), comparable outbreaks may avoid detection due to the sporadic nature of patient infections, scattering of cases throughout the hospital, and changing antibiotic susceptibility profiles. Furthermore, because of budget constraints, surveillance programs focusing on 'high-risk' isolates carrying select resistance genes (e.g. ESBLs, carbapenemases) (***World Health Organization, 2017***; ***Hong et al., 2015***; ***Del Barrio-Tofiño et al., 2020***) would also fail to detect this ST-621 outbreak clone. Indeed, while the ST-621 lineage was the focus of several studies in the late 2000s, these examined a *bla*$_{IMP-13}$ carrying epidemic clone which spread throughout Europe and

has since been sporadically detected in South America and Southeast Asia (*Mereuță et al., 2007*; *Fournier et al., 2012*; *Santella et al., 2010*; *Teo et al., 2021*). To our knowledge, this IMP-carrying strain has not been reported in the USA to date, and the outbreak in Facility A is the result of a distinct MDR clone lacking the carbapenemase.

Similar to the US MHS, healthcare networks worldwide that already benefit from prospective WGS surveillance programs are reporting large numbers of unrecognized outbreaks (*Parcell et al., 2018*). However, two resources rarely available in other settings were key to this investigation: a genome repository of isolates from the preceding decade and the ability to conduct prospective environmental sampling. The former was integral to date the presumed origin of the outbreak to the late 1990s, soon after the hospital opened, and to trace the evolution and separate spread of two subclones throughout floors and wards. The latter revealed that, more than 20 years later, reservoirs of both subclones persist in sink drains from patient rooms throughout the facility. This supports previous evidence implicating these sites as major reservoirs for *P. aeruginosa* in the clinic (*Kerr and Snelling, 2009*; *Kotay et al., 2017*). Decontamination of drains can be extremely challenging due to the limited penetration of disinfectants, the lack of access (e.g. to perform scrubbing), recolonization (e.g. disposal of contaminated patient specimens), or retrograde growth from p-traps (*Kerr and Snelling, 2009*; *Smith and Hunter, 2008*; *Kotay et al., 2017*). Furthermore, sinks with shallow basins and gooseneck faucets directing water straight into the drain, similar to those in Facility A (*Figure 4A*), have been linked to increased backsplash onto nearby surfaces and medical equipment (*Hota et al., 2009*).

Besides environmental contamination, reservoirs within patients likely contributed to the spread and longevity of the outbreak. In particular, long-term infections, documented for a significant fraction of patients, provided a recurring source of the epidemic clone. It is noteworthy that 5 of the 6 patients with long-term (>1 year) infections (and 9 out of 11 if reduced to 6 months) were carriers of subclone SC2. Compared to SC1, all but one SC2 isolates carried a truncated WbpX glycosyltransferase, an essential component of the common polysaccharide antigen (CPA) biosynthesis (*Lam et al., 2011*). While this mutation could result in increased immune evasion, the rough LPS phenotype recurrently observed in vivo (in particular in CF patients) usually results from a functional CPA but the lack of O-specific antigen (*Jurado-Martín et al., 2021*; *Hu et al., 2017*). In further evidence for a progression toward a host-adapted lifestyle, a convergence of mutations in genes involved in alginate production, quorum-sensing deficiency, loss of motility, and decreased protease secretion, all phenotypes associated with chronic infections (*Marvig et al., 2015*), was observed throughout the 10 years of sampling.

Considering its role in long-term infections and the distinctive association with patients and sinks in the Tower Building of Facility A, the origin and emergence of subclone SC2 is of particular importance. Although obfuscated by the limited sampling, a plausible scenario (supported by BEAST2 inferences) would be that a patient with an initial infection acquired during a stay in the Main Building before the SC1-2 split (ca. 2004) proceeded to shed the epidemic strain in the newly opened (2012) Tower Building during successive visits. Importantly, the Main and Tower Buildings of Facility A have distinct plumbing systems, providing the necessary ecological separation (minus inter-hospital patient transfers) for the further spread and divergence of the two sub-clones. Patient 27 is just such an example, but the lack of early sampling (2000–2010) and incomplete collection thereafter precludes a definitive answer. Indeed, other scenarios cannot be discounted, with a recent study showing sinks in a newly built ICU were already contaminated with an outbreak clone of *P. aeruginosa* before the arrival of the first patients (*Sukhum et al., 2022*).

One of the most notable features of this outbreak was the evolution of antibiotic resistance over time, with the emergence of resistance to first (cephalosporin), second (carbapenems), and third (colistin) line treatments. The potent R504C substitution in PBP3 was selected in each subclone, and in addition to facilitating cephalosporin resistance, has also been linked to ceftazidime/avibactam resistance (*López-Causapé et al., 2017*). While no patient prescription data is available from Facility A, it can be speculated that the high prevalence of cephalosporin resistance in early outbreak isolates led to increased carbapenem use, which in turn selected for the many independently evolved OprD mutants, one of the most common mechanisms for carbapenem resistance in *P. aeruginosa* (*Jurado-Martín et al., 2021*). Finally, albeit limited to a single patient, the emergence of colistin resistance, through a well-characterized mechanism, is a reminder that the threat of extensively drug-resistant *P. aeruginosa* is only a prescription and a few mutations away (*Chambers and Sauer, 2013*).

Some limitations were noted for this study and complicated the genomic inferences. First, although the criteria remained the same, the partial and inconsistent sampling of clinical *P. aeruginosa* isolates prevented from capturing the exact magnitude of the outbreak through the years, and sampling bias cannot be ruled out. Second, detailed epidemiological data for the patients (date of admission(s), bed location, reason for sampling, medications prescribed, etc.) were unavailable. As a result, possible patient overlaps were only inferred from the location and culture date of the isolates, and many were likely not detected. Third, fixed thresholds (i.e. genetic relatedness, time between isolates collection, and shared hospital location) were applied to predict a possible origin of infection. Acknowledging the other limitations, these were designed to predict the most likely cases of direct patient-to-patient transmission, when an overlap was identified, or environment-to-patient transmission, when a patient overlap was ruled out. Because of conservative criteria, the origins of most cases remained undetermined, and an alternative to fixed threshold, using TransPhylo (*Didelot et al., 2021*), is being explored in a follow-up study. This approach uses a probabilistic model to predict whom infected whom in outbreak scenarios and reconstruct transmission trees (*Didelot et al., 2021*). Fitting our dataset, TransPhylo was recently extended to allow the use of multiple genomes per host (*Carson et al., 2024*) and to incorporate epidemiological data into the analysis (*Carson et al., 2025*).

The data generated during this study has resulted in various ongoing interventions (e.g. closing sinks, replacing tubing, using foaming detergents *Jones et al., 2020*) at Facility A. Sampling and sequencing, in near real-time, of all *P. aeruginosa* clinical isolates (and not just MDR) also commenced in 2021. Notably, as of May 2025, no new ST-621 patient cases have been reported in over 15 months, and the unbiased contemporary data using all *P. aeruginosa* from patients at this facility confirms the spread has slowed, with just four cases identified in 2022–2024. Similar to Facility A, the roll-out of routine WGS surveillance in the clinic has the potential to improve patient care and prevent some of the estimated 136 million hospital-associated drug-resistant infections per year globally (*Balasubramanian et al., 2023*).

## Materials and methods

### Data collection and antibiotic susceptibility testing

As mandated by the US National Action Plan for Combating Antibiotic-Resistant Bacteria (CARB), the MRSN collects and analyzes clinically relevant MDR organisms across the US MHS and around the world in collaboration with the Global Emerging Infections Surveillance (GEIS) Branch. All isolates received by the MRSN underwent WGS (single colony from a pure culture), and criteria for isolate collection were as follows. Between 2011 and 2020, all MDR isolates (resistance to ≥3 antibiotic families) were requested from all US facilities, but compliance varied between years and facilities. For Facility A, starting in 2021, all clinical *P. aeruginosa* isolates (MDR or not) were requested.

As a result of this sampling, from 2011 to 2020, 5129 MDR *P. aeruginosa* isolates were collected from 71 facilities and sent to the MRSN. Of these, 1511 were collected from Facility A, a large tertiary US military hospital located in the continental United States. In total, 253 isolates belonged to the ST-621 outbreak clone.

Antibiotic susceptibility testing (AST) was performed in a College of American Pathologists (CAP)-accredited lab at Facility A using a Vitek 2 (card GN AST 71 and GN ID; bioMérieux, NC, USA). For isolates without MICs, AST was performed in the MRSN CAP-accredited clinical lab as previously described (*Galac et al., 2020*).

### WGS and bioinformatic analysis

Genomic DNA was extracted and sequenced via Illumina MiSeq or NextSeq benchtop sequencer (Illumina, Inc, San Diego, CA, USA) as previously described (*Galac et al., 2020*). Long-read sequencing was performed using PacBio RS II (Pacific Biosciences of California, Inc, Menlo Park, CA, USA). Kraken 2 was used to identify isolate species and check for contamination (*Wood et al., 2019*). Short-read sequencing data were trimmed for adapter sequence content and quality using bbduk (*Bushnell, 2025*). De novo assembly was performed using Newbler v2.9 (*Silva et al., 2013*). Minimum thresholds for contig size and coverage were set at 200 bp and 49.5×, respectively. Long-read sequencing data were assembled using HGAP 3.0 in the SMRT Analysis portal. In silico MLST was performed using the scheme developed by *Curran et al., 2004*. Antimicrobial resistance genes were annotated

using a combination of AMRFinderPlus and ARIBA (*Feldgarden et al., 2019*; *Hunt et al., 2017*). The genomes of all 253 ST-621 isolates have been deposited in the National Center for Biotechnology Information under BioProject PRJNA852179.

## cgMLST, gene annotation, SNP calling, and phylogenetic analysis

cgMLST were determined in SeqSphere+ (Ridom, Germany) using the cgMLST scheme developed by *Curran et al., 2004*, and a 90% cutoff (3960 of 4400 genes). SNP calling was performed with Snippy v.4.4.5 (https://github.com/tseemann/snippy) (*Seemann, 2024*) using error-corrected (Pilon v1.23) (*Walker et al., 2014*) and annotated (Prokka v1.14.6) (*Seemann, 2014*) draft assembly, with isolate MRSN 4605 as the reference (chosen because it was both collected early in the outbreak and had low root-to-tip divergence). Type strain PAO1 was used as the basis for annotation (*Stover et al., 2000*). The core SNP alignment was filtered for recombination using Gubbins v2.3.1 (*Croucher et al., 2015*). An SNP-based phylogeny was created by inferring a maximum-likelihood tree with RaxML-NG v0.9.0 (*Kozlov et al., 2019*) using GTR+G (50 parsimony, 50 random). The phylogeny was visualized in iTOL (*Letunic and Bork, 2021*).

Gene annotation and clustering showed 8754 total genes in the pangenome of the outbreak. The core genome (present in ≥99% of isolates) was comprised of 5674 genes, with a shell genome (defined as genes found in ≥95% of isolates) of 297 genes, and a cloud genome (≥15% of isolates) of 530 genes.

## Curation and analysis of recurring mutations in outbreak isolates

Genome-wide analysis of NSY and LOF mutations in all outbreak isolates showed a total of 629 mutated sites (257 singletons and 372 sites shared by ≥2 isolates) in 488 distinct genes (*Supplementary file 5*). For variant sites identified in >99% of outbreak isolates, individual nucleotide-based sequence alignments were performed using a wild-type allele from the PAO1 genome (https://www.pseudo-monas.com/). This allowed the identification of sites for which the MRSN 4605 reference genome was carrying the mutation and not the remaining outbreak isolates. These were curated (e.g. entry was modified to reflect the true mutation and its presence/absence in all outbreak isolates) manually and indicated as such in *Supplementary file 5*.

## SNP distances between isolates from different patients

SNP distances were obtained from the alignment by using snp-dist (https://github.com/tseemann/snp-dists) (*Seemann et al., 2021*). To create violin plots, the complete all-vs-all dataset was limited, based on five sets of criteria, to pairs of isolates that (1) originated from different patients; (2) originated from different patients, and were collected in the same month; (3) originated from different patients and were collected in the same ward; (4) originated from different patients, were collected in the same month, and the same ward; or (5) originated from within the same patient (*Figure 2E*).

Primary isolates were designated as originating from a patient-to-patient transmission if they were part of a pair from different patients, were collected in the same ward, were separated by ≤10 SNPs (to account for the observed within-host variability, *Figure 2E*), and were collected ≤31 days apart (as a proxy for a possible patient overlap). Primary isolates were conservatively designated as environmental if they were part of a pair from different patients, were collected in the same ward, were collected >365 days apart (as a proxy for unlikely patient overlap), and were separated by a number of SNPs ≤3x the number of years they were separated to account for the 2.987E-7 substitutions/site/year from the BEAST analysis. This conservative threshold was chosen to favor high specificity as this inference relied solely on clinical isolates (i.e. the identical environmental strain in the patient-environment-patient chain was not sampled). For these clinical isolates to have acquired no/very little mutation in that much time, no/low replication is expected and, although unsampled, we propose this most likely happened on hospital surfaces.

In each of these cases, direct transmission or transmission mediated by a healthcare worker was not distinguished as no sampling was available from the later. All other primary isolates were designated as undetermined.

## Bayesian evolutionary phylogenetic analysis

To evaluate the strength of the temporal signal, TempEst v1.5.3 was utilized to visualize the relationship between root-to-tip genetic distances for samples with known collection dates (*Rambaut et al.,*

*2016*). To date internal nodes of interest on the phylogeny, Bayesian phylogenetic inference was performed using BEAST2 v2.6.5 on a recombination-free alignment, removing samples with uncertain collection dates, and accounting for constant sites with beast2_constsites (https://github.com/andersgs/beast2_constsites) (*da Silva, 2019*; *Bouckaert et al., 2014*). The HKY substitution model was selected based on the evaluation of all possible substitution models in bModelTest v1.2.1 (*Bouckaert and Drummond, 2017*). The random clock model was selected based on support by the marginal likelihood value using the Nested Sampling package v1.1.0 (*Russel et al., 2019*). BEAST2 was run under a coalescent constant population model with a Markov chain Monte Carlo length of $1 \times 10^8$ sampling every $5 \times 10^3$ steps. Analyses were repeated five times to confirm consistency between the obtained posterior distributions. Parameter estimates were computed using Tracer v1.7.1. Posterior trees were combined with LogCombiner and summarized in TreeAnnotator after a 50% burn-in. The final MCC target tree was visualized and annotated using iTOL (https://itol.embl.de/) (*Letunic and Bork, 2021*).

## Acknowledgements

The authors are thankful to all the staff of the MRSN and the clinical microbiology laboratory of the MRSN MTF network. Additionally, we would like to thank Dr. Madeline Galac for sharing insight on phylogenetic analysis using parsimony and Dr. Xavier Didelot for helpful discussions in predicting patient transmission events during bacterial outbreaks. The manuscript has been reviewed by the Walter Reed Army Institute of Research, and there is no objection to its presentation. The views expressed herein are those of the author(s) and do not necessarily reflect the official policy or position of the Defense Health Agency, Brooke Army Medical Center, the Department of Defense, nor any agencies under the US Government. This study was funded by the US Army Medical Command and the Defense Medical Research and Development Program. In addition, this study was partly funded by the Armed Forces Health Surveillance Division (AFHSD), Global Emerging Infections Surveillance (GEIS) Branch ProMIS P0098_22_WR. The funders have no role in the decision to publish or the preparation of this article.

## Additional information

### Funding

| Funder | Grant reference number | Author |
|---|---|---|
| United States Army Medical Command | | Jason W Bennett Patrick T Mc Gann |
| Defense Medical Research and Development Program | | Jason W Bennett Patrick T Mc Gann |
| Armed Forces Health Surveillance Division - The Global Emerging Infections Surveillance program | ProMIS P0098_22_WR | Jason W Bennett Patrick T Mc Gann Francois Lebreton Melissa J Martin |

The funders had no role in study design, data collection and interpretation, or the decision to submit the work for publication.

### Author contributions

William Stribling, Data curation, Formal analysis, Investigation, Visualization, Methodology, Writing – original draft; Lindsey R Hall, Casey Harless, Katelyn V Bartlett, Brendan T Jones, Data curation, Formal analysis, Investigation, Visualization, Methodology; Aubrey Powell, Resources, Data curation, Investigation, Methodology; Melissa J Martin, Data curation, Formal analysis, Supervision, Funding acquisition, Validation, Investigation, Visualization, Writing - review and editing; Brendan W Corey, Jason Stam, Data curation, Formal analysis, Validation, Investigation, Visualization, Methodology; Erik Snesrud, Conceptualization, Data curation, Investigation; Ana Ong, Rosslyn Maybank, Lan N Preston, Katherine F Lane, Resources, Data curation, Validation, Investigation, Methodology; Bernadette Thompson, Lynn M Young, Alice E Barsoumian, Ana Elizabeth Markelz, John L Kiley,

Resources, Data curation, Validation, Investigation; Yoon I Kwak, Resources, Data curation, Investigation, Project administration; Robert J Cybulski, Resources, Validation, Investigation, Project administration; Jason W Bennett, Resources, Data curation, Supervision, Funding acquisition, Validation, Investigation, Project administration, Writing - review and editing; Patrick T Mc Gann, Conceptualization, Resources, Data curation, Supervision, Funding acquisition, Validation, Investigation, Project administration, Writing - review and editing; Francois Lebreton, Conceptualization, Data curation, Formal analysis, Supervision, Validation, Investigation, Visualization, Writing – original draft, Project administration

### Author ORCIDs
Lindsey R Hall ⓘ https://orcid.org/0000-0002-6012-9899
Brendan T Jones ⓘ https://orcid.org/0000-0001-5106-2795
Francois Lebreton ⓘ https://orcid.org/0000-0002-7157-5026

Reviewer #1 (Public review): https://doi.org/10.7554/eLife.93181.3.sa1
Reviewer #2 (Public review): https://doi.org/10.7554/eLife.93181.3.sa2
Reviewer #3 (Public review): https://doi.org/10.7554/eLife.93181.3.sa3
Author response https://doi.org/10.7554/eLife.93181.3.sa4

## Additional files

### Supplementary files
Supplementary file 1. Isolate metadata.

Supplementary file 2. All-vs-all SNP comparison.

Supplementary file 3. Environmental isolates.

Supplementary file 4. Antibiotic susceptibility testing.

Supplementary file 5. Prediction of variants in outbreak isolates.

Supplementary file 6. Resistome reference intrinsic alleles.

Supplementary file 7. Mutations in intrinsic genes associated with antibiotic resistance.

Supplementary file 8. Mutations in functional groups or pathways.

MDAR checklist

### Data availability
Both genomic assemblies and raw sequencing data of all isolates analyzed in this study are publicly available in the NCBI database under the BioProject number PRJNA852179.

The following dataset was generated:

| Author(s) | Year | Dataset title | Dataset URL | Database and Identifier |
|---|---|---|---|---|
| Stribling W, Mc Gann PT, Lebreton F | 2022 | Tracking the origins, spread and evolution of a MDR *P. aeruginosa* epidemic clone over ten years in a single hospital | https://www.ncbi.nlm.nih.gov/bioproject/PRJNA852179/ | NCBI BioProject, PRJNA852179 |

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
