## [Editor Report · eLife Assessment]

This **important** work presents an example of how genomic data can be used to improve understanding of an ongoing, long-term bacterial outbreak in a hospital with an application to multi-drug resistant *Pseudomonas aeruginosa*, and will be of interest to researchers concerned with the spread of drug-resistant bacteria in hospital settings. The **convincing** genomic analyses highlight the value of routine surveillance of patients and environmental sampling and show how such data can help in dating the origin of the outbreak and in characterising the epidemic lineages. These findings highlight the importance of understanding environmental factors contributing to the transmission of *P. aeruginosa* for guiding and tailoring infection control efforts.

---

## [Referee Report · Reviewer #1 (Public review)]

Summary:

This is a manuscript describing outbreaks of *Pseudomonas aeruginosa* ST 621 in a facility in the US using genomic data. The authors identified and analysed 254 *P. aeruginosa* ST 621 isolates collected from a facility from 2011 to 2020. The authors described the relatedness of the isolates across different locations, specimen types (sources), and sampling years. Two concurrently emerged subclones were identified from the 254 isolates. The authors predicted that the most recent common ancestor for the isolates can be dated back to approximately 1999 after the opening of the main building of the facility in 1996. Then the authors grouped the 254 isolates into two categories: (1) patient-to-patient; or (2) environment-to-patient using SNP thresholds and known epidemiological links. Finally, the authors described the changes of resistance gene profiles, virulence genes, cell wall biogenesis and signaling pathway genes of the isolates over the sampling years.

Strengths:

The major strength of this study is the utilisation of genomic data to comprehensively describe the characteristics of a long-term *Pseudomonas aeruginosa* ST 621 outbreak in a facility. This fills the data gap of a clone that could be clinically important but easily missed from microbiology data alone.

Weaknesses:

As the authors highlighted in the Discussion section, a limitation of this study is that there is potential sampling bias due to partial sampling of clinical *P. aeruginosa* isolates. However, the work is still important to showcase the potential benefits of applying genomic sequencing techniques to support infection prevention controls in hospital settings. The limitation on potential sampling bias could inspire further work to explore an optimal clinical isolate sampling framework for genomic analyses to support outbreak investigation. The other limitation that the authors have highlighted in the Discussion session is the lack of epidemiology data to support the interpretation of the inferred patient-to-patient and environment-to-patient transmissions, which emphasised the importance of metadata to complement genomic data analysis in outbreak investigation for future studies.

Impact of the work:

First, the work adds to the growing evidence implicating sinks as long-term reservoirs for important MDR pathogens, with direct infection control implications. Moreover, the work could potentially motivate investments in generating and integrating genomic data into routine surveillance. The comprehensive descriptions of the *Pseudomonas aeruginosa* ST 621 clones outbreak is a great example to demonstrate how genomic data can provide additional information about long-term outbreaks that otherwise could not be detected using microbiology data alone. Moreover, identifying the changes in resistance genes and virulence genes over time would not be possible without genomic data. Finally, this work provided additional evidence for the existence of long-term persistence of *Pseudomonas aeruginosa* ST 621 clones, which likely occur in other similar settings.

Comments on revisions:

The paper would be further strengthened from an additional timeline indicating when routine surveillance was introduced and examples of actions or changes guided by the surveillance data that resulted in decrease in ST 621 transmission. This additional information would be useful to support the final statement in the Abstract suggesting "Since initial identification, extensive infection control efforts guided by routine, near real- time surveillance have proved successful at slowing transmission."

---

## [Referee Report · Reviewer #2 (Public review)]

Summary:

The authors present a report of a large *Pseudomonas aeruginosa* hospital outbreak affecting more than 80 patients with first sampling dates in 2011 that stretched over more than 10 years and was only identified through genomic surveillance in 2020. The outbreak strain was assigned to the sequence type 621, an ST that has been associated with carpabapenem resistance across the globe. Ongoing transmission coincided with both increasing resistance without acquisition of carbapenemase genes as well as convergence of mutations towards a host-adapted lifestyle.

Strengths:

The convincing genomic analyses indicate spread throughout the hospital since the beginning of the century and provide important benchmark findings for future comparison

The sampling was based on all organisms sent to the Multidrug resistant Organism Repository and Surveillance Network across the U.S. Military Health System.

Using sequencing data from patient and environmental samples for phylogenetic and transmission analyses as well as determining recurring mutations in outbreak isolates allows for insights into the evolution of potentially harmful pathogens with the ultimate aim of reducing their spread in hospitals.

Weaknesses:

The epidemiological information was limited and the sampling methodology was inconsistent, thus complicating inference of exact transmission routes. Epidemiological data relevant for this analysis include information on the reason for sampling, patient admission and discharge data and underlying frequency of sampling and sampling results in relation to patient turnover.

Comments on revisions:

Thank you for the careful revision and consideration of my comments.

I am pleased to confirm that all my concerns have been comprehensively addressed.

The changes and additions made have resolved my initial feedback, and I see no need to alter my evaluation.

---

## [Referee Report · Reviewer #3 (Public review)]

Summary:

This paper by Stribling and colleagues sheds light on a decade-long *P. aeruginosa* outbreak of the high-risk lineage ST-621 in a US Military hospital. The origins of the outbreak date back to the late 90s and it was mainly caused by two distinct subclones SC1 and SC2. The data of this outbreak showed the emergence of antibiotic resistance to cephalosporin, carbapenems and colistin over time highlighting the emerging risk of extensively resistant infections due to *P. aeruginosa* and the need for ongoing surveillance.

Strengths:

This study, overall, is well constructed and clearly written. Since detailed information on floor plans of the building and transfers between facilities was available, the authors were able to show that these two subclones emerged in two separate buildings of the hospital. The authors support their conclusions with prospective environmental sampling in 2021 and 2022 and link the role of persistent environmental contamination to sustaining nosocomial transmission. Information on resistance genes in repeat isolates for the same patients allowed the authors to detect the emergence of resistance within patients. The conclusions have broader implications for infection control at other facilities. In particular, the paper highlights the value of real-time surveillance and environmental sampling in slowing nosocomial transmission of *P. aeruginosa*.

Weaknesses:

My major concern is that the authors used fixed thresholds and definitions to classify the origin of an infection. As such, they were not able to give uncertainty measures around transmission routes nor quantify the relative contribution of persistent environmental contamination vs patient-to-patient transmission. The latter would allow the authors to quantify the impact of certain interventions. In addition, these results represent a specific US military facility and the transmission patterns might be specific to that facility. The study also lacked any data on antibiotic use that could have been used to relate to and discuss the temporal trends of antimicrobial resistance.

Comments on revisions:

The authors have addressed my concerns adequately in the revised manuscript.

---

## [Author Response]

The following is the authors’ response to the original reviews.

**Reviewer #1 (Public Review):**
Summary:This is a manuscript describing outbreaks of *Pseudomonas aeruginosa* ST 621 in a facility in the US using genomic data. The authors identified and analysed 254 *P. aeruginosa* ST 621 isolates collected from a facility from 2011 to 2020. The authors described the relatedness of the isolates across different locations, specimen types (sources), and sampling years. Two concurrently emerged subclones were identified from the 254 isolates. The authors predicted that the most recent common ancestor for the isolates can be dated back to approximately 1999 after the opening of the main building of the facility in 1996. Then the authors grouped the 254 isolates into two categories: (1) patient-to-patient; or (2) environment-to-patient using SNP thresholds and known epidemiological links. Finally, the authors described the changes in resistance gene profiles, virulence genes, cell wall biogenesis, and signaling pathway genes of the isolates over the sampling years.Strengths:The major strength of this study is the utilisation of genomic data to comprehensively describe the characteristics of a long-term *Pseudomonas aeruginosa* ST 621 outbreak in a facility. This fills the data gap of a clone that could be clinically important but easily missed from microbiology data alone.Weaknesses:The work would further benefit from a more detailed discussion on the limitations due to the lack of data on patient clinical information, ward movement, and swabs collected from healthcare workers to verify the transmission of *Pseudomonas aeruginosa* ST 621, including potential healthcare worker to patient transmission, patient-to-patient transmission, patient-to-environment transmission, and environment-to-patient transmission. For instance, the definition given in the manuscript for patient-to-patient transmission could not rule out the possibility of the existence of a shared contaminated environment. Equally, as patients were not routinely swabbed, unobserved carriers of *Pseudomonas aeruginosa* ST 621 could not be identified and the possibility of misclassifying the environment-to-patient transmissions could not be ruled out. Moreover, reporting of changes in rates of resistance to imipenem and cefepime could be improved by showing the exact p-values (perhaps with three decimal places) rather than dichotomising the value at 0.05. By doing so, readers could interpret the strength of the evidence of changes.Impact of the work:First, the work adds to the growing evidence implicating sinks as long-term reservoirs for important MDR pathogens, with direct infection control implications. Moreover, the work could potentially motivate investments in generating and integrating genomic data into routine surveillance. The comprehensive descriptions of the *Pseudomonas aeruginosa* ST 621 clones outbreak is a great example to demonstrate how genomic data can provide additional information about long-term outbreaks that otherwise could not be detected using microbiology data alone. Moreover, identifying the changes in resistance genes and virulence genes over time would not be possible without genomic data. Finally, this work provided additional evidence for the existence of long-term persistence of *Pseudomonas aeruginosa* ST 621 clones, which likely occur in other similar settings.

We thank the reviewer for their thorough evaluation of our work, and for the suggested improvements. A main goal of this study was to show that integrating routine wgs in the clinic was a game changer for infection control efforts. We appreciate this aspect was highlighted as a strength by this reviewer. While some of the weaknesses identified are inherent to the data (or lack thereof) available for this study, we have revised the manuscript to include a detailed discussion on limitations (sampling, thresholds of genetic relatedness, definition and categories etc.) that could influence the genomic inferences. We also provided exact p-values for the changes in rates of resistance, as requested. Finally, we have positively answered all the specific recommendations suggested by the reviewer and modified the manuscript accordingly.

**Reviewer #2 (Public Review):**
Summary:The authors present a report of a large *Pseudomonas aeruginosa* hospital outbreak affecting more than 80 patients with first sampling dates in 2011 that stretched over more than 10 years and was only identified through genomic surveillance in 2020. The outbreak strain was assigned to the sequence type 621, an ST that has been associated with carpabapenem resistance across the globe. Ongoing transmission coincided with both increasing resistance without acquisition of carbapenemase genes as well as the convergence of mutations towards a host-adapted lifestyle.Strengths:The convincing genomic analyses indicate spread throughout the hospital since the beginning of the century and provide important benchmark findings for future comparison.The sampling was based on all organisms sent to the Multidrug-resistant Organism Repository and Surveillance Network across the U.S. Military Health System.Using sequencing data from patient and environmental samples for phylogenetic and transmission analyses as well as determining recurring mutations in outbreak isolates allows for insights into the evolution of potentially harmful pathogens with the ultimate aim of reducing their spread in hospitals.Weaknesses:The epidemiological information was limited and the sampling methodology was inconsistent, thus complicating the inference of exact transmission routes. Epidemiological data relevant to this analysis include information on the reason for sampling, patient admission and discharge data, and underlying frequency of sampling and sampling results in relation to patient turnover.

We thank the reviewer for their thoughtful feedback on our manuscript and for highlighting the quality of the genomic analyses. We agree that the lack of patient epi data (e.g. date of admission and discharge) and the inconsistent sampling through the years are limitations of this study. We have revised the manuscript to acknowledge these limitations and discuss how not having this data complicates the inference of exact transmission routes. Finally, we have positively answered all the specific recommendations suggested by the reviewer and modified the manuscript accordingly.

**Reviewer #3 (Public Review):**
Summary:This paper by Stribling and colleagues sheds light on a decade-long *P. aeruginosa* outbreak of the high-risk lineage ST-621 in a US Military hospital. The origins of the outbreak date back to the late 90s and it was mainly caused by two distinct subclones SC1 and SC2. The data of this outbreak showed the emergence of antibiotic resistance to cephalosporin, carbapenems, and colistin over time highlighting the emerging risk of extensively resistant infections due to *P. aeruginosa* and the need for ongoing surveillance.Strengths:This study overall is well constructed and clearly written. Since detailed information on floor plans of the building and transfers between facilities was available, the authors were able to show that these two subclones emerged in two separate buildings of the hospital. The authors support their conclusions with prospective environmental sampling in 2021 and 2022 and link the role of persistent environmental contamination to sustaining nosocomial transmission. Information on resistance genes in repeat isolates for the same patients allowed the authors to detect the emergence of resistance within patients. The conclusions have broader implications for infection control at other facilities. In particular, the paper highlights the value of real-time surveillance and environmental sampling in slowing nosocomial transmission of *P. aeruginosa*.Weaknesses:My major concern is that the authors used fixed thresholds and definitions to classify the origin of an infection. As such, they were not able to give uncertainty measures around transmission routes nor quantify the relative contribution of persistent environmental contamination vs patient-to-patient transmission. The latter would allow the authors to quantify the impact of certain interventions. In addition, these results represent a specific US military facility and the transmission patterns might be specific to that facility. The study also lacked any data on antibiotic use that could have been used to relate to and discuss the temporal trends of antimicrobial resistance.

We thank the reviewer for their evaluation of our work and for highlighting the broad implications of our findings regarding the application of real-time surveillance to suppress nosocomial transmission. We agree with the reviewer that fixed thresholds and definitions are imperfect to classify the origin of an infection. The design of this study (e.g. inconsistent sampling through time) was not conducive to provide a comprehensive/quantitative measurement of transmission routes. Thus, we decided to apply conservative thresholds of genetic relatedness and strict conditions (e.g. time between isolate collection, shared hospital location etc.) to favor specificity as our goal was simply to establish that cases of environmentto-patient transmission did happen. In the absence of a truth set, we have not performed sensitivity analysis, but we are conducting a follow-up study to compare inferences from MCMC models to our original fixed-thresholds predictions. This limitation is now discussed in the revised manuscript. Finally, we have positively answered all the specific recommendations suggested by the reviewer and modified the manuscript accordingly including the addition of Figure S3.

**Reviewer #1 (Recommendations For The Authors):**
The definitions used on lines 391-396 are necessarily somewhat arbitrary, but it would be helpful to have a little bit more justification for the choices made, particularly for the definition of environmental involving the "3x the number of years they were separated". It seems a little hard to square this with the more relaxed 10 SNP cutoff for a patient-to-patient designation. Are there reasons for thinking SNP differences associated with environmental transmission should be smaller than for patient-to-patient, or is the aim here just to set the bar higher for assuming an environmental source? Because these definitions are quite arbitrary, there could also be some value in exploring the sensitivity of the results to these assumptions.

Thank you. We agree with the reviewers that SNP thresholds, albeit necessarily, are arbitrary and that more discussion/justification was needed to put the genomic inferences in context. We have revised the manuscript to indicate that: 1/ the 10 SNP cutoff for a patient-to-patient designation was set to account for the known evolution rate of *P. aeruginosa* (inferred by BEAST at 2.987E-7 subs/site/year in this study and similar to previous estimates PMID: 24039595) and the observed within host variability (now displayed in revised Fig. 1E). We note that this SNP distance was not sufficient and that an epi link (patients on the same ward at the same time) needed to be established. 2/ the environment-to-patient definition was indeed set to be most conservative (nearly identical isolates in two patients from the same ward with no known temporal overlap for > 365 days). This was indeed done to favor high specificity as this inference relied solely on clinical isolates (i.e. the identical environmental strain in the patientenvironment-patient chain was not sampled). For these clinical isolates to have acquired no/very little mutation in that much time, no/low replication is expected and, although unsampled, we propose this most likely happened on hospital surfaces.

While the term "core genome" should be familiar to most readers, "shell genome" and "cloud genome" are less widely known, and an explanation of what these terms mean here would be helpful.

Thank you. We have revised the manuscript to define the core, shell, and cloud genomes as genes sets found in ≥ 99%, ≥ 95% and ≥ 15% of isolates, respectively.

In the first paragraph of the discussion, it could be added that in many cases for clinically important Gram negatives short read sequencing alone will fail to detect transmission events as outbreaks can be driven by plasmid spread with only very limited clonal spread (see, for example, https://www.nature.com/articles/s41564-021-00879-y)

Thank you. We agree this is an important/emerging aspect of surveillance. However, the goal of this discussion point was to explain why such a large outbreak was missed prior to implementing WGS (short read) surveillance. We feel that discussing “plasmid outbreaks” (which is not at play here, and relatively rare in *P. aeruginosa* compared to the Enterobacteriaceae) and the need for long read will distract from the narrative.

line 599 What does "Mock" mean here? Would it be more accurate to say it is a simplified floor plan?

Thank you. “Mock” was changed to “simplified”

IPAC abbreviation is only used once - spelling it out in full would increase readability.

Revised manuscript was edited as suggested.

MHS is only used twice.

Revised manuscript was edited to spell out Military Health System

Line 364: full stop missing.

Revised manuscript was edited as suggested.

Line 401: Bayesian rather than bayesian.

Revised manuscript was edited as suggested.

**Reviewer #2 (Recommendations For The Authors):**
Thank you for giving me the opportunity to review this interesting manuscript.The conclusions of this paper are mostly well supported by the data presented, but epidemiological information was limited and the sampling methodology was inconsistent, thus complicating inference of exact transmission routes.Major issues:What was the baseline frequency of clinical and/or screening samples of *Pseudomonas aeruginosa* at the hospital? Neither Figure 1D nor Table S1 allows for differentiating between clinical and screening samples. Most isolates were cultured from clinical materials, and there is no information about the patients' length of stay and their respective sampling dates. Is there any possibility of finding out whether the samples were collected for clinical or screening purposes? Would it be possible to include the patients' admission data to determine whether the strains were imported into the hospital or related to a previous stay, e.g. among known carriers? Also, the issue of sampling dates vs. patient stay on the ward should be addressed, as there may be an overlap in patients' stay on the ward but no overlap in terms of sampling dates or even missing samples (missing links).

We have revised the manuscript to address this important point: (i) 16 isolates were from surveillance swabs and are labelled “Surveillance” in Table S1. The remaining 237 were clinical isolates; (ii) unfortunately, because the sampling was done under a public health surveillance framework, we do not have access to historical patient data (admission/discharge date, wards, rooms, etc.) and we can not calculate length of stay or better identify patient overlap. These limitations are now acknowledged in the discussion of the revised manuscript.

In order to evaluate the extent of the outbreak, more epidemiological data would be useful What is the size of the hospital, what is the average patient turnover, and what is the average length of stay in ICU and non-ICU? Is there any specialization besides the military label?

We have revised the manuscript to indicate that facility A is 425-bed medical center and is the only Level 1 trauma center in the Military Health System. Unfortunately, the data to calculate length of stay, throughout the years, in ICU and non-ICU, was not available to us. This limitation is now also acknowledged in the discussion.

Perhaps the authors could attempwt to discuss the extent to which large outbreaks like these may be considered as part of unavoidable evolutionary processes within the hospital microbiome as opposed to accumulation and transmission of potentially harmful genes/clones, and differentiate between the putative community spread without any epidemiological links on the one hand, and hospital outbreaks that could be targeted by local infection prevention activities on the other hand.,

We respectfully disagree with the suggestion that this large outbreak “may be considered as part of unavoidable evolutionary processes within the hospital microbiome” and should be opposed to “transmission of potentially harmful genes/clones”. As a matter of fact, our data showed that infection control staff at Facility A responded with multiple interventions, including closing sinks, replacing tubing, and using foaming detergents. This resulted in slowing the spread of the ST621 outbreak with just 3 cases identified in 2022, 0 cases in 2023 and 1 case in 2024. This is now discussed in the revised manuscript.

Page 5, lines 88-92 lines 101-104. It seems as if the outbreak was identified only by the means of genomic surveillance. This raises questions as to the rationale for sampling and sequencing, especially prior to 2020. Considering 11 cases per year between 2011 and 2016, one could assume such an outbreak would have been noticed without sequencing data.

The MRSN was created in 2010, in response to the outbreak of MDR Acinetobacter baumannii in US military personnel returning from Iraq and Afghanistan. Between 2011 and 2017, the MRSN collected MDR isolates (mandate for all MDR ESKAPE but compliance varied between years and facilities) from across the Military Health System and, for select isolates (e.g. high-risk isolates carrying ESBLs or carbapenemases) performed molecular typing by PFGE. In 2017 the MRSN started to perform whole genome sequencing of its entire repository. In 2020, a routine prospective sequencing service was started and first detected the ST621 outbreak. A retrospective analysis of historical isolate genomes (2011-2019) identified additional cases. The first paragraph of the discussion lists possible factors to explain why the ST621 escaped detection by traditional approaches. We believe 11 cases per year is not a strong signal when stratified by month, wards, or both, especially for a clone lacking a carbapenemase and without a remarkable antibiotic susceptibility profile.

Did the infection control personnel suspect transmission? If yes, was the sampling and submission of samples to the MRSN adapted based on the epidemiologic findings?

The ST621 outbreak was unsuspected before the initial genomic detection in 2020. Until that point, MDR isolates only (Magiorakos et al PMID: 21793988) were collected but compliance was variable through time. Quickly thereafter (starting in 2021), complete sampling of all clinical *P. aeruginosa* (MDR or not) from Facility A was started. The manuscript was revised to clarify those details of the sampling strategy.

Is there any information about how many environmental sites were sampled without evidence of ST621 / screening samples were cultured without evidence of *Pseudomonas aeruginosa*?

For patient isolates, only 16 isolates were from surveillance swabs. The remaining 237 were clinical isolates. No denominator data was available to calculate *P. aeruginosa* and ST-621 positivity rate in surveillance swabs throughout the time period. For environmental isolates, a total of 159 swabs were taken from 55 distinct locations in 8 wards/units including the ER. This data is now included in the revised manuscript. However, a complete analysis of these swabs (positivity rate for ESKAPE pathogens, *P. aeruginosa*, per ward/floor/room, per swab type (sink drain, bed rail etc.) etc.) is beyond the scope of this study and is being performed as a follow up investigation.

Page 5 lines 89 and 39 Figure S1B. Please describe how the allelic distance for the cluster threshold was selected.

As indicated in the legend of Figure S1B, no thresholds were applied. All ST621 isolates ever sequenced by the MRSN were included. All except 3 isolates shared between 023 cgMLST allelic differences. The remaining 3 were distant by 88-89 allelic differences. The text was revised to clarify this point.

Page 5 lines 99-100. Could the authors please provide some distribution measures (e.g. IQR).

Done as requested. The revised manuscript now reads “…of just 38 single nucleotide polymorphisms (SNPs), and an IQR of 19 (Fig. 1A, Table S1).”

Page 5 line 102. Could the authors please provide some distribution measures (e.g. IQR).

Please see above. A chart was created and is now included as Fig. S2.

Page 6 line 107 and page 34 figure 1c. In the text it is stated that isolates were collected in 27 wards, the figure 1C depicts 26 wards and n/a.

Thank you for spotting this inconsistency. This has been fixed in the revised manuscript.

Page 6 lines 117-118. Samples collected in the emergency room would imply samples collected on admission, already addressed previously. Did the authors investigate a potential import into the hospital from community reservoirs or were all these isolates collected among patients who had been previously admitted to the hospital and/or tested positive for the outbreak strain?

We agree that samples collected in the ER imply samples collected on admission. Of the 29 ER isolates only 9 (31%) were primary isolates (first detection in a new patient) which suggests a majority were from returning patients at Facility A. Because the sampling was done under a public health surveillance framework, we do not have access to historical patient data (admission/discharge date, wards, rooms, etc.) to investigate/confirm that these 9 patients had previous visits at Facility A. This point is now discussed in the revised manuscript.

Page 6 line 128. This could also represent increased selective pressure. However, according to Table S1, the 28 isolates collected in 2011 (the number does not match with Figure 1D) were from many different wards, thus indicating earlier spread throughout the hospital.

Yes, we agree. Please note that table S1 lists all isolates for 2011 whereas Figure 1D focuses on primary (first isolate from each patients) only.

Page 7 line 133. Both Figure 2 and the discussion section, page 13 line 296 suggest the year 2005 instead of 2004?

Thank you for catching this typographical error. This was corrected to 2004 in the revised manuscript.

Figure 1E. The figure should also depict intra-patient diversity for comparison.

Thank you for this great suggestion. We have revised Figure 1E accordingly.

Page 7, lines 146-147 Could the authors attempt explaining the upper part of the bimodal peaks?

This is an all-vs-all SNP analysis for all inter-patient isolates. For each isolates all distances to other isolates are reported, not only the smallest. The upper peaks represent comparisons to isolates from a different outbreak subclone (SC1 vs SC2).

Page 7, line 150 This is a very small number considering the extent of the outbreak and suggests a large number of missing links. Or does this rather imply continuous import and evolution over time that does not necessarily represent transmission within the hospital?

We believe all cases were due to transmission happening within the hospital. Based on conservative thresholds (genetic relatedness and epi link, or lack thereof) the precise origin from another patient (n=10) or a contaminated surface (n=12) can be inferred. For the remaining 60 patients, with the available sampling, the conditions we chose are not met and we simply do not conclude whether a direct patient-to-patient or an environmental origin was more likely.

Page 8 line 155. What does the temporal overlap refer to - sampling date versus patient's stay on the ward? Please specify.

The temporal overlap was investigated from sampling dates, as dates of patient admission/discharged were not available.

Page 8, line 157: What does primary/serial isolate mean - first and follow-up samples of ST621 per patient?

Yes. Primary isolate is used to designate the first isolate from a patient. Serial isolates designate follow-up samples of ST621.

Page 8 line 165: Table S3 and Figure 3 only refer to environmental samples from three wards. Ward 20 rooms 2 and 18 as well as ward 1 rooms 1 and 6 were hotspots - is there any information on the specific infection control/disinfection measures? Addressed in discussion page 12, lines 273-275, but no information on what was actually done.

The manuscript was revised to indicate the precise disinfection measures that were taken. A follow-up study is ongoing to assess long-term efficacy and monitor possible retrograde growth from previously contaminated sinks.

Page 8 line 175: Evaluation of change in resistance fraction over time - There may have been a selection bias with an inconsistent number of strains sequenced per year.

Yes, incomplete sampling and possible selection bias are now listed with other limitations of this study in the discussion of the revised manuscript.

Page 9 line 183: The referral to Table S1 is unclear, I could not find the number and the specific isolates selected for long-read sequencing.

Thank you. This has been added to the revised Table S1.

Page 10 lines 217-225 and Figure 4C: Perhaps it is possible to better align what is written in the text and the caption of the figure. The caption does not clarify that only one patient develops colistin resistance (what was the reason to include the other patients?).

Thank you. We have revised the text and the caption of the figure to clarify that only isolates from one patient developed colistin resistance. The isolates from the other patients on Fig. 4C are shown to provide context and accurately map the emergence of the PhoQE77fs mutation.

Page 10, lines 228-229 and Table S5: How is it possible to identify those 64 genes in Table S5?

We have revised Table S5 to facilitate the identification of the 64 genes with ≥ 2 independently acquired mutations (excluding SYN). Specifically, we have added column E labeled “Counts independent mutations per locus (excluding SYN)”. A total of 205 rows (in this table each row is a variant) have a value ≥ 2 and these represent 64 genes (upon deduplication of locus tags).

Page 13, lines 280-281: Where is the information on chronic infection presented? Serial cultures would not necessarily mean chronic infection.

Authors response: Yes, we agree this was not the appropriate characterization and this was revised to ‘long-term’ infections.

Page 14 line 306: Emergence of colistin resistance in a single patient, correct?

Yes. This was further clarified in the text.

Page 14 lines 315-320: This should go to the results section. In particular disinfection, closing, and replacing of tubing should be mentioned in the results section in reference to the results presented in Table S3.

Thank you. We have considered this suggestion and have decided to leave this discussion as the closing paragraph of this publication. A follow-up study is ongoing to assess long-term efficacy of these interventions on the ST-621 bur also other outbreak clones at Facility A.

MethodsPage 15 lines 330-333: Perhaps it is possible to avoid redundancy.

Thank you. We have revised the text accordingly.

Page 15 lines 341: Information on which isolates were subjected to long-read sequencing is missing.

Thank you. This has been added to the revised Table S1.

Page 16 line 345: Was there a particular reason why Newbler was chosen?

No. At the time Newbler was the default assembler built in the MRSN bacterial genome analysis pipeline and QC processes.

Page 16, line 357-358: What was the rationale for selecting this isolate as reference genome?

This isolate was chosen because it was collected early in the outbreak and phylogenetic analysis revealed it had low root to tip divergence.

Page 16 line 361: Why 310 isolates, if only 253 were assigned to the outbreak clone and only a subset of those were collected in facility A?

This was a typographical error that has corrected (it now reads “…set of 253 isolates.”) in the revised manuscript.

Page 17 lines 387-395: What is the reason that intra-patient diversity was not included in the set of criteria for SNP distances?

The observed within host variability (now displayed in revised Fig. 1E) was taken into consideration when setting SNP thresholds for categorizing patient-to-patient transmission or environment-to-patient event. This is now clarified in the revised manuscript.

Page 17 line 392: How was the threshold of <= 10 SNPs determined?

The 10 SNP cutoff to infer a patient-to-patient transmission event was set to account for the known evolution rate of *P. aeruginosa* (inferred by BEAST at 2.987E-7 subs/site/year in this study, and similar to previous estimates PMID: 24039595) and the observed within host variability (now displayed in revised Fig. 1E). We note that this SNP distance was not sufficient and that an epi link (patients on the same ward within the same month) needed to be established.

Page 17 line 395 and Figure 2: What was the assumed average mutation rate per genome per year?

Thank you. The mean substitution rate inferred by BEAST was 2.987E-7 similar to estimate from previous studies on *P. aeruginosa* outbreaks (e.g. PMID: 24039595).

**Reviewer #3 (Recommendations For The Authors):**
Please find (line-by-line comments) on each section of the manuscript below:IntroductionLine 86: I am wondering why the authors state ">28 facilities" instead of the exact number of facilities from which these lineages were recovered.

Thank you. Manuscript was revised to provide the exact number of facilities. It now reads “…recovered from 37 and 28 facilities, respectively.”

MethodsIt's not clear to me which criteria were used for collecting these isolates (both prospective and retrospective). I understand that some of the data are described in more detail in Lebreton et al but I did not find the specific criteria for the collection of the isolates and I imagine that these might differ if different facilities. Would it be possible to comment on that and add a short paragraph in the Methods section?

Thank you. This lack of clarity was also raised by other reviewers, and we have revised the manuscript to indicate that: 1/MDR isolates only (Magiorakos et al PMID: 21793988) were collected from 2011-2020 with the same criteria for all facilities although compliance was variable through time and between facilities; and 2/ starting in 2021 all *P. aeruginosa* isolates, irrespective of their susceptibility profile, were collected from Facility A

The data comes from a US Military hospital. Is this related to the US Veterans Affairs Healthcare system? Is there more detailed information about the demographics of the patient population?

Facility A is part of the Military Health System (MHS) which provides care for active service members and their families. This is distinct from the US Veterans Affairs Healthcare system. Only limited patient data was accessible to us as this study was done as part of our public health surveillance activities. Patient age (avg. 57.2 +/- 21.0) and gender (ratio male/female 1.7) are provided in the revised manuscript.

Line 384ff: The origin of infection was inferred based on the SNP threshold and epidemiological links. However, recombination events can complicate the interpretation of SNP data. Have the authors attempted to account for this?

Thank you. We agree that recombination events can complicate the interpretation of SNP data. We used Gubbins v2.3.1 to filter out recombination from the core SNP alignment, as indicated in the revised manuscript.

The authors' definition of environment-to-patient transmission seems conservative (nearly identical strain and no known temporal overlap for > 365 days). Have the authors changed the threshold, performed sensitivity analyses, and tested how this would affect their results?

Indeed, acknowledging that fixed thresholds have limitations in their ability to accurately predict the origin of infections, we took a conservative approach to favor specificity as our goal was simply to establish that cases of environment-to-patient transmission did happen. In the absence of a truth set, we have not performed sensitivity analysis, but we are conducting a follow-up study to compare inferences from MCMC models to our original predictions. This limitation is now discussed in the revised manuscript.

The authors don't seem to incorporate the role of healthcare workers in the transmission process. Could they comment on this? I am assuming that environment-to-patient transmission could either be directly from the environment to the patient or via a healthcare worker. I think it's fine to make simplifying assumptions here but it would be great if this was explicitly described.

Thank you for this suggestion. We have not sampled the hands of healthcare workers in this study. As a result, the reviewer is correct to say that we made the simplifying assumption that healthcare workers would be possible intermediates in either environment-topatient or patient-to-patient transmissions, as previously described by others (PMID: 8452949). This limitation is now discussed in the revised manuscript.

Page 5, line 100: What does "all vs all" mean? Based on the supplement, I assume it's the pairwise distance and then averaged across all of those. It would improve the readability of the manuscript if the authors could briefly define this term and then maybe refer to Table S1.

Thank you. We have created Fig.S2 and revised the manuscript to state that ST-621 isolates from facility A belonged to the same outbreak clone with a distance (averaged all vs all pairwise comparison) of just 38 single nucleotide polymorphisms (SNPs), and an IQR of 19 (Fig. S2, Table S1).

Figure 1D: It would be interesting to see additional figures in the supplement on the percentage of sequenced isolates per year and whether it varies across the different sources/sites. Is there any information on which isolates were chosen for sequencing?

Lack of clarity in the sampling/sequencing scheme was raised by multiple reviewers and we have provided a thorough response to earlier comments. We also have revised the material and methods section accordingly. Finally, we have created Fig. S3 to show the percentage of sequenced isolates per year across different sources/sites, as suggested by the reviewer. No noticeable patterns were observed.

It seems like only a subset of all clinical isolates were sequenced. Would it be possible that SC2 was present already earlier but not picked up until a certain date?

Although all isolates received by the MRSN were sequenced, compliance varied through time so it is true that not all clinical isolates were sequenced between 2011-2019. As such, we fully agree with this hypothesis and discuss this possibility as BEAST analysis placed the origin of SC2 in 2004 while the first detection of an SC2 isolate was in December 2012. This limitation is now discussed in the revised manuscript.

Could the authors elaborate on whether the isolates resulted from single-colony picks? Is it possible that the different absence of a subclone is due to the fact that they picked only a colony?

Yes, the isolates resulted from single-colony picks except when the presence of different colony morphologies was noted. In the latter, representative isolates for each colony morphologies were processed. We have revised the methods to make that clear.

Figure 2: It is difficult to see which nodes belong to which patient due to the small font size. I wonder if it was possible to color the nodes for each patient, to make it more readable.

We tried coloring the nodes but with > 60 distinct patients/colors we decided it did not improve clarity. We have revised figure 2 to increase the font size.

Page 7-8, lines 154-155: Did the authors check whether there were isolates of the same strain (that were found in the environment) present in other patients elsewhere in the ward?

Yes. In rare cases, we observed virtually genetically identical isolates from two patients collected in different wards. Because we only have access to clinical isolate data (collected from patient X in ward Y) and do not have access to patient data (admission/discharge date, wards, rooms, etc.), we do not know but cannot exclude that patients overlap in a room prior to the sampling of their *P. aeruginosa* isolates. We designed our fixed thresholds to be conservative. As a result, in this analysis, these cases are labelled as “undetermined”.

Page 8: Do the authors have any information on antibiotic use during this timeframe? From the discussion, it seems like there is no patient-level prescription data. Is there any data on overall trends? How were trends in antibiotic use correlated with trends in antibiotic resistance?

Unfortunately, patient-level prescription data (or any other data not linked to the bacterial specimens) was not accessible to us as this study was done as part of our public health surveillance activities.

To infer the origin of infection, the authors used a static method with fixed thresholds and definitions. This study does not provide any uncertainty with their estimates. Maybe the authors could add a sentence in the discussion section that MCMC methods to infer transmission trees incorporating WGS could provide these estimates. These methods have not been applied to PA a lot but two examples where MCMC methods have been used without WGS (though the definition of environmental contamination may differ between these studies and this study).
https://doi.org/10.1186/s13756-022-01095-x

https://doi.org/10.1371/journal.pcbi.1006697

Thank you for this great suggestion. We have revised the manuscript to include a discussion on the limitations of fixed thresholds to infer transmission chains/origins, and to discuss existing alternatives including MCMC methods.

Line 322-323: This sentence is a bit vague since not all of these HAI are due to *P. aeruginosa*. I would suggest citing a number that is specific to PA.

Thank you. While our paper shows a particular example of protracted *P. aeruginosa* outbreak, the roll-out of routine WGS surveillance in the clinic will help prevent hospital-associated drug-resistant infections for more than this species. We believe that broadening the scope in the last sentence of the manuscript is important and we decline to revise as suggested.